# Stability Control of Quadruped Robot Based on Active State Adjustment

**DOI:** 10.3390/biomimetics8010112

**Published:** 2023-03-09

**Authors:** Sai Gu, Fei Meng, Botao Liu, Zhihao Zhang, Nengxiang Sun, Maosen Wang

**Affiliations:** 1Intelligent Robotics Institute, School of Mechatronical Engineering, Beijing Institute of Technology, Beijing 100081, China; 2Beijing Advanced Innovation Center for Intelligent Robots and Systems, Beijing Institute of Technology, Beijing 100081, China

**Keywords:** balance control, active state adjustment, disturbance recovery, active environment adaptation

## Abstract

The quadruped robot has a strong motion performance and broad application prospects in practical applications. However, during the movement of the quadruped robot, it is easy to be affected by external disturbance and environmental changes, which makes it unable to achieve the ideal effect movement. Therefore, it is very important for the quadruped robot to adjust actively according to its own state detection. This paper proposes an active state adjustment control method based on its own state, which can realize disturbance recovery and active environment adaptation. Firstly, the controller is designed according to the physical model of the quadruped robot, and the foot forces are optimized using the quadratic program (QP) method. Then, the disturbance compensation method based on dynamic analysis is studied and combined with the controller itself. At the same time, according to the law of biological movement, the movement process of the quadruped robot is actively adjusted according to the different movement environment, so that it can adapt to various complex environments. Finally, it is verified in a simulation environment and quadruped robot prototype. The results show that the quadruped robot has a strong active disturbance recovery ability and active environment adaptability.

## 1. Introduction

In recent years, the research on the quadruped robot has been gradually oriented to practical scene applications [1,2,3]. In practical applications, the main problems faced by quadruped robots are unknown external disturbances and environmental changes. Therefore, at present, the research of the quadruped robot focuses on how to improve the motion performance, disturbance recovery ability and active environment adaptation of the quadruped robot [4]. Among them, the most representative achievements are a series of robots developed by Boston Dynamics, such as BigDog [5], LittleDog [6], WildCat and SpotMini. They are absolutely technologically advanced in motion flexibility, load capacity, stability, etc. In addition, IIT developed the high-performance hydraulic quadruped robot “HyQ” series [7]. MIT developed the flexible and high-speed “Cheetah” series robot [8,9,10]. ETH developed the robot “StarlETH” with the leg flexibility control [1,11], and the complex terrain adaptive robot “ANYmal” with an autonomous learning ability [12]. In order to have wider application scenarios for quadruped robots [13], such as transport supplies and rescue tasks in complex environments [14], it is very important to study the active disturbance recovery ability and active complex environment adaptation ability of the quadruped robot.

When the quadruped robot body is disturbed by the external disturbance [15], the motion state of the quadruped robot will change in a short time, which makes the robot obtain the unexpected velocity and acceleration, so that the robot will enter the unstable state or even fall down [16]. In order to maintain stability after an external disturbance, the quadruped robot needs to identify and estimate the external shock, and carry out targeted whole body balance control. In the process of dealing with the disturbance, the real-time attitude of the robot can be calculated based on the information of the sensor [17]. Then, based on the dynamics model, the expected lateral acceleration of the inverse dynamics controller can be calculated by the lateral displacement and lateral velocity of the quadruped robot to restore the balance [18]. This method takes few factors into consideration and cannot stabilize the robot quickly. Another method is to design a nonlinear observer to estimate the external force exerted on the robot [19], and to realize the stable walking of the quadruped robot by compensating the external force disturbance through the sliding mode controller [20]. The external force obtained by this method is not accurate, and it is adjusted passively, which cannot ensure the accurate control adjustment of the robot. It can be observed that in order to realize the autonomous disturbance resistance of the robot, it is necessary to accurately respond to the external disturbance and comprehensively consider all kinds of instability conditions.

In addition, the stability of the quadruped robot motion will be greatly affected by the change of environment. If we want to make the quadruped robot have a better motion effect, we need to make a real-time adjustment according to the environment of the robot [21,22]. In previous research, most quadruped robots respond to environmental changes by enhancing the robustness of their own controllers [10], such as the model predictive control and whole-body control [23]. The model design of this method is complicated, and has not been adjusted for different environments. Some robots use a gait planning method to move in different types of complex environments [24,25,26], which requires visual assistance and real-time calculation and greatly reduces the efficiency of motion [27]. Inspired by this, on the premise of ensuring the motion effect and efficiency, properly simplifying the model and making active adjustments for different complex environments will have a better applicability.

To enhance the active disturbance recovery ability and active complex environment adaptability of the quadruped robot, it is very important for the quadruped robot to adjust actively according to its own state detection. This paper designs a controller based on the real model of the quadruped robot, which mainly includes the virtual model control and GRF optimization solution based on the QP method [28,29]. Then, based on the dynamic analysis of the robot, an active disturbance compensation method, which can be combined with the controller, is proposed. At the same time, according to the biological movement law, the control method of the quadruped robot is actively adjusted for different complex environments [30]. The method has good universality, reliability and accuracy. The main contributions of this paper are as follows:

(1) According to the stance phase and swing phase of the quadruped robot and based on the dynamic analysis of the robot, the posture compensation, velocity compensation and air swing leg compensation are carried out, respectively, for the robot body.

(2) Based on the biological motion law, the control method of the quadruped robot’s motion process is adjusted according to different motion environments. An active environment adaption control method is proposed to achieve the balance control of the robot by actively adapting the body posture and adjusting the swing leg trajectory without additional perceptual or visual information.

The rest of this paper is organized as follows. Section 2 introduces the modeling and control framework of quadruped robot. Section 3 introduces disturbance compensation control methods for different aspects. Section 4 introduces the active environment adaptive control method of the quadruped robot in different environments. The simulation and experimental results are discussed in Section 5. The conclusions of this study and the direction of future works are discussed in Section 6.

## 2. Locomotion Control Based on GRF Optimization

The quadruped robot is a kind of multi-DOF and comprehensive system; its control system is very complex. During various environmental movements of the robot, the trajectory generator generates the state instructions of the target, and the state estimator computes the current state variables in real time, which are jointly input into the controller. The controller is divided into the control stance phase robot under the trunk and the body of the supporting leg controller and controls the motion of the swinging leg controller. The controller calculates the foot force corresponding to the four legs and calculates it into the joint torque, which is converted into the current command and sent to the driver, so as to realize the direct force control of the robot with a high dynamic motion. The robot’s control framework is shown in Figure 1:

For the overall movement of the robot, most of the mass of the robot is concentrated on the trunk of the robot, and the leg rod is composed of a lightweight carbon rod with a negligible mass. Therefore, for the simplicity of the model, in this paper, the robot model is a single rigid body model. In addition, GRF is the only external force acting on the foot; thus, the control model can be expressed as follows: (1)$$\left[\begin{array}{ccc} I_{3} & \cdots & I_{3}\\ ({\mathrm{Toe}}_{1}-p_{com})\times & \cdots & ({\mathrm{Toe}}_{4}-p_{com})\times \end{array}\right]\left[\begin{array}{c} F_{grf,1}\\ \cdots \\ F_{grf,4} \end{array}\right]=\left[\begin{array}{c} m({\ddot{p}}_{com}+g)\\ I_{g}{\dot{\omega }}_{b} \end{array}\right]$$
where *m* and $I_{g}$ are the robot mass and inertia, g is the gravity, ${\mathrm{Toe}}_{i}$ is the foot position of each leg of the robot, $p_{com}$ is the position of the center of mass of the robot, and ${\dot{\omega }}_{b}$ is the angular velocity of trunk rotation.

A virtual model controller is added to the six degrees of freedom of the trunk at the center of mass:(2)$$\left[\begin{array}{c} F_{c,d}\\ \tau_{c,d} \end{array}\right]=\left[\begin{array}{c} K_{p}^{p}\left(p_{com}^{d}-p_{com}\right)+K_{d}^{p}\left({\dot{p}}_{com}^{d}-{\dot{p}}_{com}\right)\\ K_{p}^{\theta }\left(\theta_{com}^{d}-\theta_{com}\right)+K_{d}^{\theta }\left({\dot{\theta }}_{com}^{d}-{\dot{\theta }}_{com}\right) \end{array}\right]$$
where $F_{c,d}$ and $\tau_{c,d}$ are the virtual force and torque acting on the center of mass, $p_{com}^{d}$ and $p_{com}$ are the desired and actual position of the center of mass, and $\theta_{com}^{d}$ and $\theta_{com}$ are the desired and actual angular of the center of mass.

Next, the calculated virtual force at the center of mass is transformed into a quadratic optimization problem by the QP solver and distributed to each foot, and the real-time update is realized in the control period. The force distribution problem can be written in the following form: (3)$$\begin{array}{cc} f^{*} & =min\;{\left(Af-b_{d}\right)}^{T}S\left(Af-b_{d}\right)+f^{T}Wf\\ & s.t.\;Cf\leq d \end{array}$$
where f is the reaction force of the foot on the ground, A is the foot position, C is the friction coefficient matrix, $b_{d}$ is the virtual force acting on the center of mass of the trunk calculated according to Equation (Equation 2), and S and W are the weight matrices used to adjust the optimization results of the virtual force and torque. In order to avoid the relative sliding between the foot and the ground, a friction cone is added to limit Cf≤d to keep the optimized force within the maximum and minimum amplitude.

After the foot force $F_{grf,i}$ is obtained, the joint torque $\tau_{i}$ can be obtained according to the mapping relationship between the joint torque $\tau_{i}$ and foot force $F_{grf,i}$: (4)$$\tau_{i}=J^{T}F_{grf,i}$$
where $\tau_{i}$ is the torque to be provided by each joint, and *J* is the Jacobian matrix of the robot leg.

## 3. Disturbance Compensation Control

The quadruped robot should take the initiative to detect its own state in real time during the movement, and adjust its own state in time after being subjected to an unknown disturbance. The motion process of the actual robot can be divided into two stages: stance phase and swing phase. The stance leg is the leg that touches the ground to provide support for the body, and the swing leg is the leg that swings forward to provide forward speed for the body. Different compensation control strategies are required for different stages. The compensator for the stance phase includes the posture compensation according to the actual posture of the trunk and the velocity compensation according to the trunk velocity. The compensation of the stance leg is to increase or reduce the torque applied to the joints, while the compensation of the swing leg is to compensate for the position of the foot end landing point.

### 3.1. Trunk Posture Compensation

Because the overall mass and inertia of the robot are large, too much inclination will easily cause the robot to roll over. Therefore, we need to set a stability threshold, and when the expected value is significantly different from the actual value, $-\pi /36\leq \theta_{p}^{d}-\theta_{p}^{a}\leq \pi /36$, we need to compensate the robot attitude. As shown in the Figure 2a, when there is a horizontal tilt of the trunk, the compensation amount is calculated according to the current horizontal tilt angle of the body, so that the joint exerts the reverse torque to recover the level of the trunk.
(5)$$\tau_{pos}=-k_{p}^{pos}\left(\theta_{p}^{d}-\theta_{p}^{a}\right)-k_{d}^{pos}\left(\omega_{p}^{d}-\omega_{p}^{a}\right)$$
where $\tau_{pos}$ is the torque required for posture compensation, $k_{p}^{pos}$ and $k_{d}^{pos}$ are the control coefficient of the body tilt angle and angular velocity stabilization, $\theta_{p}^{d}$ and $\theta_{p}^{a}$ are the desired and actual roll angler of the trunk, and $\omega_{p}^{d}$ and $\omega_{p}^{a}$ are the desired and actual roll angular velocity of the trunk.

### 3.2. Trunk Velocity Compensation

When the tilt angle of the trunk does not obviously change and if there is a unexpected velocity of the trunk, it will also cause the center of gravity of the robot to move sideways, which will cause the robot to roll over. Therefore, the velocity compensation is introduced into the external pendulum joint to gradually reduce the velocity of the robot, and finally the unexpected motion trend completely disappears, as shown in the Figure 2b. The torque generated by the velocity compensation can be calculated by the following formula: (6)$$\tau_{vel}=-k_{p}^{v}v_{y}-k_{d}^{v}{\dot{v}}_{y}$$
where $\tau_{vel}$ is the torque required for the velocity compensation, $k_{p}^{v}$ and $k_{d}^{v}$ are the control coefficient of trunk velocity stabilization, $v_{y}$ and ${\dot{v}}_{y}$ are the velocity and acceleration of the trunk.

### 3.3. Air Swing Leg Compensation

When there is an unexpected velocity of the robot trunk, if the original step remains unchanged, the overall center of mass will shift out of the stable range, so that the robot can not maintain the normal motion state. The stable motion of the quadruped robot requires the center of mass to be located in the supporting range formed by the GRF (ground reaction force), so the robot needs to adjust the position of the foot point for stable support with the change of the center of mass position. The swing leg compensation provides a motion range for supporting the leg torque offset by increasing or decreasing the distance of the landing foot. The swing leg compensation requires one to determine the lateral compensation of $\theta_{1}$ and $\theta_{2}$ for the foot position according to the actual trunk inclination and lateral velocity, as shown in Figure 2c. The calculation method is as follows:(7)$$\Delta toe_{y}=\lambda_{1}L_{leg}sin\theta_{p}^{a}+\lambda_{2}\int {\dot{v}}_{y}$$
where $\Delta toe_{y}$ is the foot end position compensation for the increase or decrease in the swing leg, $\lambda_{1}$ and $\lambda_{2}$ are the compensation coefficient of the foot end position compensation of the swing leg, and $L_{leg}$ is the length of the swing leg.

## 4. Active Environment Adaptation Control

In the process of the quadruped robot movement, it is very important to adapt to the unknown changes of the external terrain in the unstructured environment. In real life, legged animals adjust the stiffness of their leg muscles according to the actions they perform and the terrain, adapting to various sports environments. Therefore, the quadruped robot needs to actively adjust its own state in real time according to the actual situation, to complete various movements.

According to the law of biological movement, the actual animal will adjust its own state according to its own state and the sensation that the foot end touches the ground. An attitude sensor is installed on the trunk of the quadruped robot to feel its own state, encoders at the joints are used to sense the spatial position of the four legs, and force sensors are installed on the sole to interact with the contact surface.

The quadruped robot uses the force sensors on the plantar to judge the ground status of each leg. The method of judgment is to determine whether the contact force exceeds the threshold. If it exceeds the threshold, it is the contact state; if it is less than the threshold, it proves that it is still in the air swing state. In addition, the angle of the ground can be judged by the values of each direction of the six-dimensional force sensors according to the geometric relationship.

### 4.1. Grassland and Rockland Road

When quadrupeds walk on grass, stones, or surfaces with a very low slope, they adjust the range of motion of their legs according to the bump of the surface. However, in this case, their trunk remains horizontal. This is because the swing of the trunk will consume more energy, and will also cause the overall shake. Therefore, when the robot moves on a slope less than 5°, the trunk should be kept horizontal to make the robot more stable. At this time, it is also necessary to adjust the stiffness and damping of the whole leg of the robot, as well as the speed of the foot end.

When the foot end contacts the raised road surface, because the foot end still has speed when contacting, the impact may cause the foot end to rebound. Therefore, when the force sensor has judged the touchdown but the foot velocity is not zero, it is necessary to reduce the stiffness of the contact leg and increase the damping of the contact leg according to the velocity in the vertical direction.
(8)$$k_{p}^{t}=\left(1-\frac{{\dot{toe}}_{z}^{a}}{{\dot{toe}}_{z}^{max}}\right)k_{p}^{sw}\;k_{d}^{t}=\left(1+\frac{{\dot{toe}}_{z}^{a}}{{\dot{toe}}_{z}^{max}}\right)k_{d}^{sw}$$
where $k_{p}^{t}$ and $k_{d}^{t}$ are the stiffness and damping of the contact leg at this time, $k_{p}^{sw}$ and $k_{d}^{sw}$ are the stiffness and damping of the swinging leg, $\dot{t}oe_{z}^{a}$ is the velocity of the foot relative to the center of mass of the quadruped robot in the z-direction, and $\dot{t}oe_{z}^{max}$ is the maximum velocity of the foot relative to the center of mass of the quadruped robot in the z-direction.

When the foot end touches the concave road surface, it may cause an instability of the robot because the foot end speed has been zero but still does not touch the ground. Therefore, when the foot tip speed is zero but the force sensor has not yet judged the touchdown, it is necessary to give the foot tip a speed in the z-direction to ensure stable contact between the foot end and the contact surface. At this time, the controller of the robot will immediately switch the state when the foot end detects a touchdown. After the switch is completed, the stiffness and damping of the leg will be changed to make it stable and not bounce:(9)$${\dot{toe_{z}}}^{a}=k_{f}{\dot{toe_{z}}}^{max}$$
where $k_{f}$ is the control coefficient of the actual velocity. To ensure the stable contact between the robot and any passive environment, the feet are made of elastic materials. The size of $k_{f}$ is related to the material of the foot.

### 4.2. Slight Slope Road

When quadrupeds are moving on a slope with a certain angle, in order to better balance their bodies, they will ensure that their trunks are parallel to the slope where they are, so as to achieve a normal gait. The robot keeps the leg length and stride unchanged, and can know the slope according to the posture sensor on the trunk. In the environment of the small slope $\theta_{s}$ (5–10°), in order to enable the robot to pass smoothly, the leg lifting height should be increased appropriately to avoid touching the ground in advance. Assume that the change height of the foot relative to the robot’s center of mass position in the vertical direction is $h_{p}$ in the plane motion and the change height is $h_{s}$ in the slope motion.
(10)$$h_{s}=h_{p}+\frac{\theta_{s}}{k_{s}}$$
where $k_{s}$ is the control coefficient of the slope. The coefficient is related to the motion space of the robot leg. The robot selects $k_{s}=5$.

At the same time, as the robot moves on a slope, the effective friction force provided by the ground becomes smaller. It needs to provide a greater force perpendicular to the slope to increase the forward friction and avoid sliding between the foot end and the slope. At this time, in order to make the robot’s feet fully contact the slope, it is necessary to increase the stiffness $k_{p}^{s}$ and damping $k_{d}^{s}$ of the legs at this time.
(11)$$k_{p}^{s}=\left(1+\frac{2\theta_{s}}{\pi }\right)k_{p}^{sw}\;k_{d}^{s}=\left(1+\frac{2\theta_{s}}{\pi }\right)k_{d}^{sw}$$

### 4.3. More Than 10° Slope Road

When a quadruped moves on a slope with a large angle (greater than 10°), it will reduce the height of its trunk to reduce the risk of falling due to its high center of mass. Therefore, the actual robot needs to reduce the height of the center of mass according to the slope of the inclined plane to achieve a more stable state during the movement process.
(12)$$H_{s}=\left(1-\frac{\theta_{s}}{4\pi }\right)H_{p}$$
where $H_{p}$ and $H_{s}$ are the height of the center of mass in the plane motion and the height of the center of mass in the inclined plane motion.

## 5. Simulation and Experiment Results

This section will conduct the simulation and experimental verification of the disturbance compensation method and passive environment adaptation that are proposed above. The robot parameters used in the simulation and experiment are completely consistent, and the specific parameters are shown in Table 1.

### 5.1. Disturbance Compensation Simulation and Experiment

#### 5.1.1. Disturbance Recovery Simulation

The simulation environment is realized by C++ and CoppeliaSim. In the simulation of the disturbance test, a ball with uniform mass is used to apply 50 N·s impulses to the body of the robot to observe the reaction state of the quadruped robot.

When the quadruped robot is disturbed, it can adjust quickly according to the disturbance compensation control. The maximum velocity of the robot during the disturbance is 1.25 m/s. When viewed from the front of the robot, the motion state of the robot in this process is shown in the Figure 3:

As shown in Figure 3, when the disturbance makes contact with the body, the disturbance begins to push the robot sideways. At this time, in order to maintain the balance of the robot, the trunk velocity compensation and air swing leg compensation have started to compensate to the controller. After that, the trunk starts to tilt significantly, and both the stance leg and swing leg have produced significant trunk posture compensation. At the end of the disturbance, the robot body velocity has reached the maximum. At this time, the robot is still moving rapidly to the right, but the velocity has been gradually reduced under the action of the compensator. With the effect of compensation, the velocity of the robot body gradually decreases to zero. In order to offset the velocity, the robot body tilts to the left, but at this time, the swing leg compensator has no part of the air swing leg compensation, but due to the existence of the inclination of the body, it begins to produce some compensations of landing points in the opposite direction. Eventually, the robot returns to its normal trot gait, the body is horizontal, and the compensator is no longer functional.

As can be observed from Figure 4a, the velocity of the robot body gradually increases after the disturbance begins and reaches the maximum at the end of the disturbance. Then, under the action of the disturbance compensation and recovery method, it gradually decreases to the state before the disturbance.

It can be observed from Figure 4b that after the disturbance began, the roll angle gradually increased along a clockwise direction. After the disturbance, the roll angle gradually decreases to the state before the disturbance because of the compensation moment. Due to the velocity, the roll angle increases further along the counterclockwise direction. Until the unexpected velocity reaches zero, the roll angle stops increasing and returns to zero in the subsequent stabilization. The pitch angle oscillates until it stabilizes.

In conclusion, in the case of large impulses (50 N·s), the robot can achieve disturbance recovery with the combined action of the stand leg and the swing leg compensator.

#### 5.1.2. Disturbance Recovery Experiment

Figure 5 shows the sequential snapshots of the quadruped robot in the stable motion state, which starts to adjust autonomously after receiving a lateral disturbance and finally stabilizes in the initial state.

As shown in Figure 6a, owing to the IMU errors, only small velocity errors exist in x- and z-directions. The external disturbance in the y-direction lasted about 0.5 s, and the maximum velocity in the y-direction reached 2 m/s. Then, the robot uses its own controller and compensation control to reach a basically stable state after 0.5 s, and there is no need to continue moving in the y-direction at this time. After about 0.4 s, a complete stability was achieved. As shown in Figure 6b, when the external disturbance lasts for about 0.5 s, the robot body roll angle reaches the maximum of 10° and the robot body pitch angle reaches the maximum of 2°. The robot body and roll angle body is recovered to the horizontal state faster by using the posture compensation.

As shown in Figure 7a, owing to the robot body velocity of y-direction and pitch angle in the whole process change little, the x-direction adjustment range of the foot end is small, up to 6 cm. As shown in Figure 7b, when the robot body begins to be impacted, the robot firstly tries to use its own controller to stabilize it, instead of stepping immediately to adjust. When the robot body roll range reached the maximum and the robot body velocity of the y-direction was still gradually increasing, the robot immediately uses its own controller and the air swing leg compensation controller proposed in 3.3 to calculate a new swing trajectory in real time to realize the leg movement, and the maximum step distance reached 22 cm. Then, in order to make the robot enter the stable state faster, the four legs of the robot will move to the forward direction of velocity before the trunk to offset the robot body velocity of the y-direction. Finally, the robot enters a stable state of standing still. The angle, velocity, and position in all directions are zero.

As shown in Figure 8a, when the robot moves in place, the GRF in the y-direction is almost zero. After the lateral disturbance, the maximum GRF in the y-direction is 75 N. As shown in Figure 8b, the robot moves with a trot gait, so the GRF in the z-direction of the two touching legs in a normal motion is about 200 N. If one of the legs touches the ground first after the disturbance, the GRF in the z-direction will reach 400 N.

### 5.2. Active Environment Adaptation Experiment

#### 5.2.1. Grassland and Rockland Experiment

Figure 9 shows the sequential snapshots of the quadruped robot stably walking on grassland and rock terrain. The quadruped robot first passed over grassland with irregular steel pipes and then passed the rock terrain. The motion is uniform and continuous throughout the process.

As shown in Figure 10a, the robot body roll and pitch angles fluctuate all the time due to slight fluctuations in the grassland and rock terrain. However, under the action of the active environment adaptive controller, the roll and pitch angles are always within the range of plus or minus 1°, around zero. As shown in Figure 10b, the GRF in the z-direction of the two contact legs in normal motion is less than 200 N. GRF remained stable and did not produce significant fluctuations on grass or stone surfaces, proving that the stiffness and damping regulation played a good control effect.

#### 5.2.2. Slight Slope Experiment

Figure 11 shows the sequential snapshots of the slight up-slope with a trot gait. As shown in Figure 12a, it took 7 s for the quadruped robot to walk from the flat to the slope, and its velocity fluctuates greatly during this transition process. Then, the robot stably walked on the slope of about 6° for about 18 s. Finally, it took 6 s to walk from the slope to the flat. As shown in Figure 12b, when the quadruped robot stepped on the slope, the actual height of the foot end from the ground increased by 2 cm from the original 7 cm to 9 cm. As shown in Figure 12c, due to the small friction coefficient and the change of the direction of friction on the slope, the GRF in the z-direction increased by about 20 N during the process from the flat to the slope in order for the robot to pass through the slope smoothly. The foot position and the first GRF changed significantly when going on slope, and then the whole process immediately remained stable without obvious fluctuations, which proved that the control method of the leg lifting height and the adjustment of stiffness damping played a good role.

#### 5.2.3. The 12° Slope Experiment

Figure 13 shows the sequential snapshots of the up-12° slope with a trot gait. As shown in Figure 14a, it took 6 s for the quadruped robot to walk from the flat to the slope, and its velocity fluctuates greatly during this transition process. After that, the robot walked on the slope with the maximum slope of 11°. As the slope was rough, it did not maintain a stable slope on the whole. The robot walked on the slope for about 21 s. Finally, it took 5 s to walk from the slope to the flat. As shown in Figure 14b, when the quadruped robot stepped on the slope, the actual height of the foot end from the ground increased by 2.5 cm from the original 7 cm to 9.5 cm. The whole process changed slightly with the change of slope, which proves that the control method of increasing the leg lifting height has been effectively verified. At the same time, due to the large slope at that time, the robot added the center of mass height adjustment control. The robot’s center of mass height (the height of the z-direction position of the foot end) is reduced by about 1 cm. As shown in Figure 14c, due to the larger slope angle, the GRF in the z-direction increased by about 30 N during the process from the flat to the slope.

### 5.3. Experimental Results Discussion

In the experiment section, the disturbance compensation control method was verified by the robot disturbance recovery experiment. Then, the robot was used to walk in different environments (such as grass, gravel, slope, etc.) to verify the active environment adaptation control method. After the completion of the experiment, the robot body velocity, angle, foot end position, and foot force were analyzed. Since the data were obtained by the position encoders, IMU after Kalman filtering, and force sensors, all the data have a small range of fluctuations. However, the overall data are in line with the experimental expectations, which proves the effectiveness of the disturbance compensation control method and the active environment adaptation control method. The quadruped robot can actively adjust according to its own state detection and has achieved good experimental results.

## 6. Conclusions

This paper proposed a method for the quadruped robot to make an active adjustment according to its own state. Firstly, according to the physical model of the quadruped robot, the controller framework based on the whole body control and QP optimization of the foot end force is completed. The active state adjustment is divided into a disturbance recovery and active environment adaptation. On this basis, the quadruped robot is vulnerable to external forces and environmental changes. This paper presents a method to quickly recover the stability of the quadruped robot after being impacted by an external force based on dynamic compensation. At the same time, the control method of quadruped robots’ motion process is adjusted according to biological motion rules and different motion environments. An active environment adaption control method is proposed to achieve the balance control of the robot by adapting the body posture and adjusting the swing leg trajectory without additional perceptual or visual information. The controller itself adjusts the foot force and swing trajectory in real time according to the environment. Although it cannot achieve a very precise adjustment control, it still has good adaptive adjustment ability and traffic capacity in various environments. The simulation environment and real robot experiment results show that the quadruped robot has a strong active disturbance recovery ability and active adaptability to various complex environments. In future work, we plan to further study the high dynamic motion of the quadruped robot, such as the ability to bound and jump, in order to improve the motion performance of the quadruped robot.

## Figures and Tables

**Figure 1 biomimetics-08-00112-f001:**
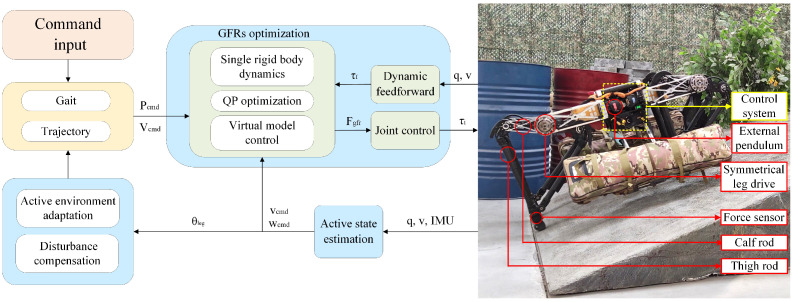
Block diagram of active state adjustment control framework.

**Figure 2 biomimetics-08-00112-f002:**
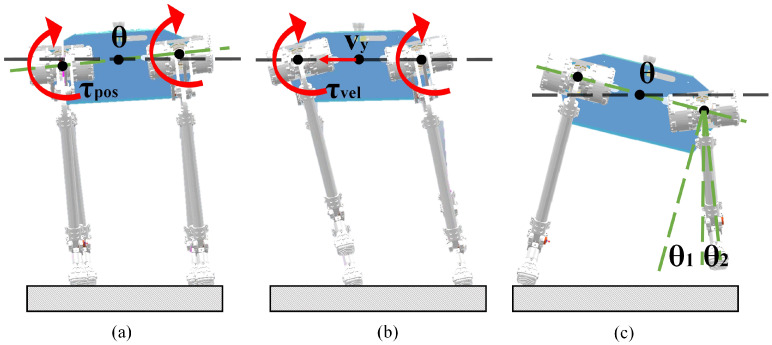
Compensation control diagram: (**a**) trunk posture compensation; (**b**)trunk velocity compensation; (**c**) air swing leg compensation.

**Figure 3 biomimetics-08-00112-f003:**
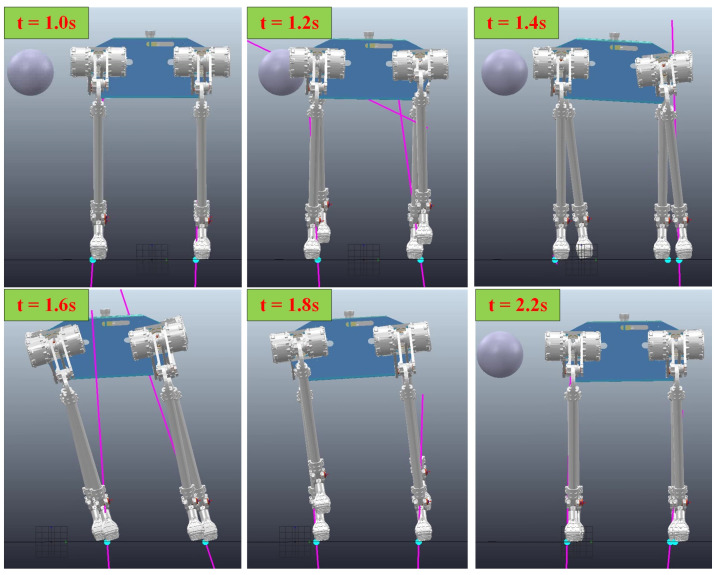
Snapshots of the disturbance recovery simulation.

**Figure 4 biomimetics-08-00112-f004:**
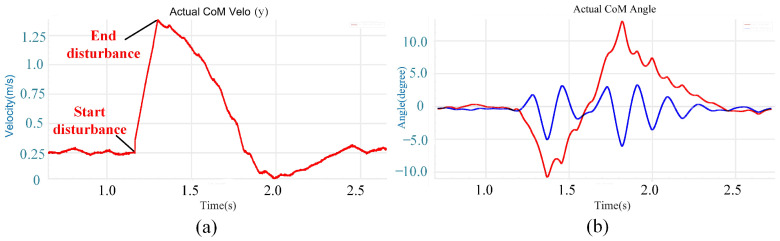
(**a**) Robot body velocity; (**b**) robot body roll and pitch angles. The red line represents the roll angle and the blue line represents the pitch angle.

**Figure 5 biomimetics-08-00112-f005:**
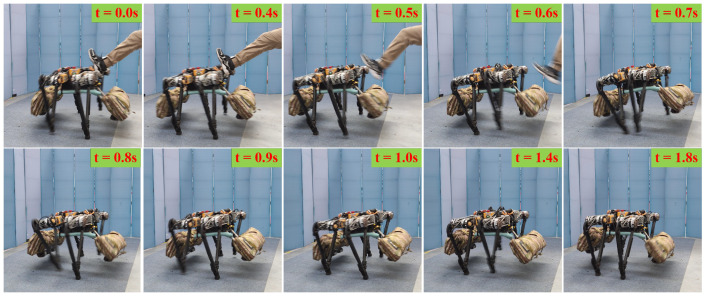
Snapshots of the disturbance recovery experiment.

**Figure 6 biomimetics-08-00112-f006:**
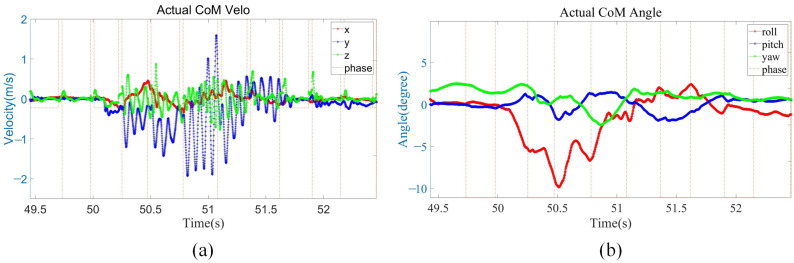
Disturbance recovery experiment data. (**a**) Robot body velocity. The red, blue, and green lines present the robot body velocity of x-, y-, and z-directions; (**b**) robot body angle. The red, blue, and green lines present the robot body roll, pitch, and yaw angles.

**Figure 7 biomimetics-08-00112-f007:**
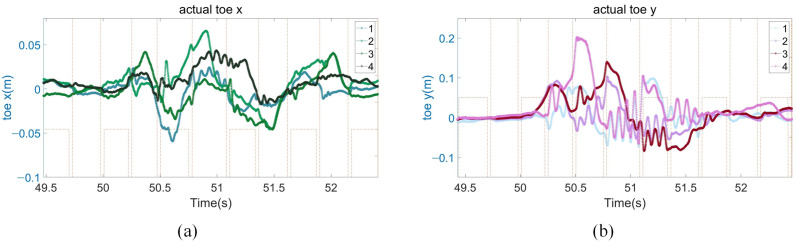
Disturbance recovery experiment data. (**a**) The x-direction position of the foot end of the four legs; (**b**) the y-direction position of the foot end of the four legs.

**Figure 8 biomimetics-08-00112-f008:**
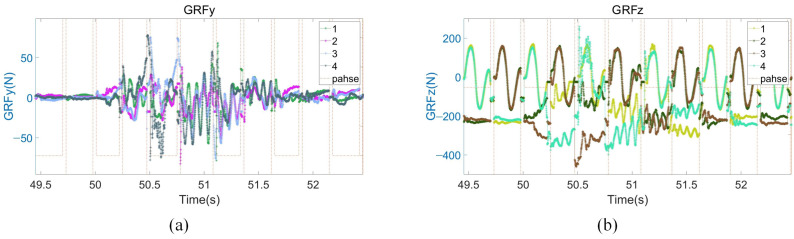
Disturbance recovery experiment data. (**a**) GRF in the y-direction; (**b**) GRF in the z-direction.

**Figure 9 biomimetics-08-00112-f009:**
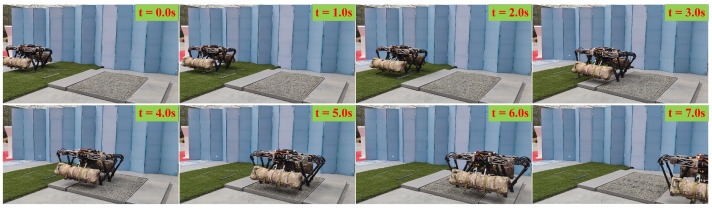
Snapshots of the walking on the grassland and rock terrain experiment.

**Figure 10 biomimetics-08-00112-f010:**
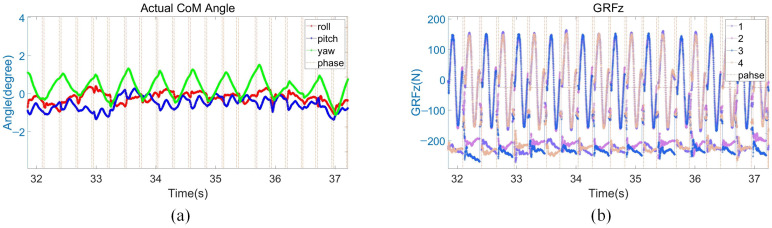
Grassland and rockland experiment data. (**a**) Robot body angle. The red, blue, and green lines present the robot body roll, pitch and yaw angles; (**b**) GRF in the z-direction.

**Figure 11 biomimetics-08-00112-f011:**
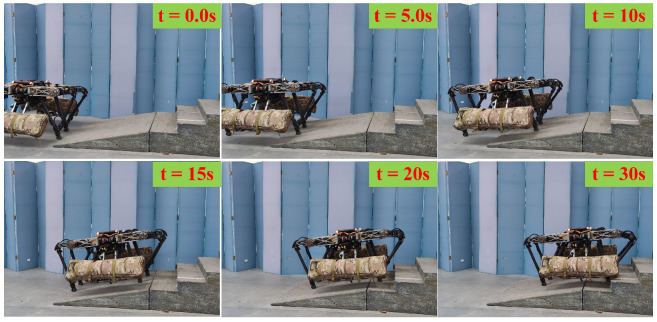
Snapshots of the slight slope experiment.

**Figure 12 biomimetics-08-00112-f012:**
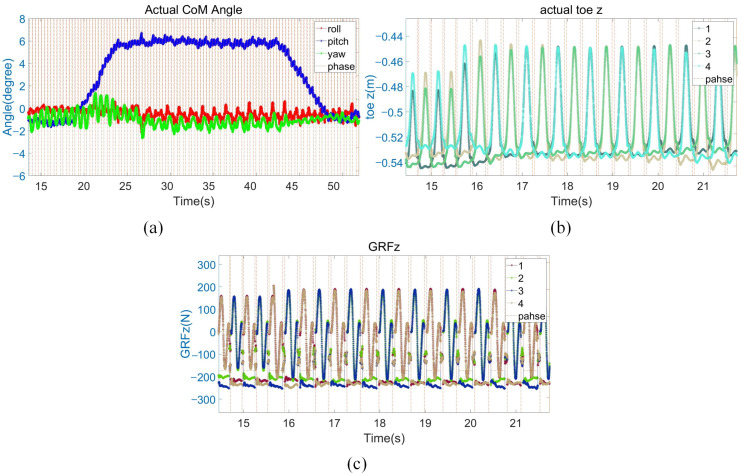
Slight slope experiment data. (**a**) Robot body angle. The red, blue, and green lines present the robot body roll, pitch, and yaw angles; (**b**) the z-direction position of the foot end of the four legs; (**c**) GRF in the z direction.

**Figure 13 biomimetics-08-00112-f013:**
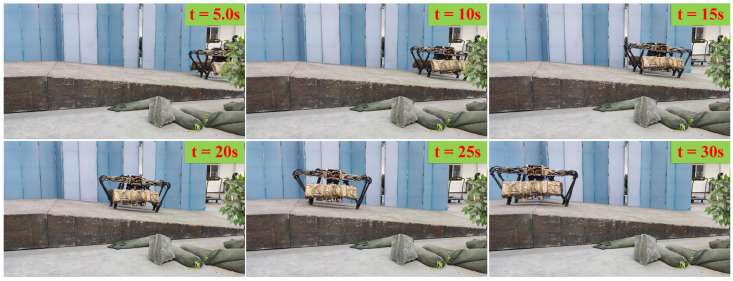
Snapshots of the 12° slope experiment .

**Figure 14 biomimetics-08-00112-f014:**
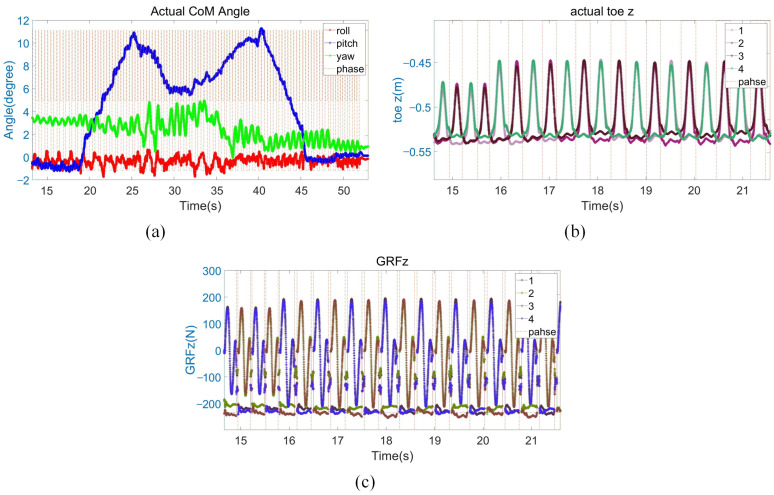
The 12° slope experiment data. (**a**) Robot body angle. The red, blue, and green lines present the robot body roll, pitch, and yaw angles; (**b**) the z-direction position of the foot end of the four legs; (**c**) GRF in the z-direction.

**Table 1 biomimetics-08-00112-t001:** Parameters of model in simulation and experiment.

Items	Value	Unit
Degrees of freedom	4 × 3	/
Total mass	40	kg
Trunk length	0.83	m
Trunk width	0.33	m
Thigh length	0.2	m
Calf length	0.55	m
Friction coefficient	0.7	/

## Data Availability

Not applicable.

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
