# Peer review of "Stability Control of Quadruped Robot Based on Active State Adjustment"

_biomimetics, 2023, doi:10.3390/biomimetics8010112_

Round 1

Reviewer 1 Report

In this paper the problem of robotic quadrupedal locomotion is tackled. The authors claim that the proposed method adapts to the environment and compensates for the disturbances, but in my opinion, only some simple PD-like controllers are proposed that are said to be combined via a QP controller (the utilization of which is not explained in the text). The presentation of the paper needs serious revision, as in its current form there are many points that cannot be understood or inferred by the reader. My detailed comments are:

1. It is not clear how and where the QP solver was utilized. The authors did not give any explanation about how the QP solver was implemented, or which QP solver they utilized. 

2. In this paper no mechanism for adjusting the control methods according to different motion environments is proposed (as claimed by the authors). On the other hand, three different PD controllers (having different control objectives) are simply proposed corresponding to three different terrains, without explaining how the robot could identify the terrain and switch between those control modes automatically. Furthermore, the rationale behind most of the controller are not explained rigorous way, but in most of the cases they are based on intuition (such as the controllers for the inclined surfaces)

3. Eq. (3) is not comprehensible if the parameters and variables are not introduced/defined. What is A? What is the difference between F_grf,i and f? What is C? What is b_d?

4. In my opinion the authors utilize the term "compensation" misleadingly, as it is utilized many times (e.g. in 3.1 and 3.2) to refer to simple PD-like controllers (without any compensation terms). Why do the authors selected to use such a term?

5. It is not clear where the control law of Eq. (7) refers to. Is \Delta toe_y  a torque, a velocity or a difference/step? Of which? Furthermore, it is not clear why this specific control law is selected. Is this control law proven to be stable?

6. In (8), the authors assume that there is a "contact leg" and a "swinging leg" (if I understood correctly). First of all, if this is true, the authors should be more clear about this and elaborate more on the assumed situation/case (e.g. a figure can be provided). Secondly, it is not clear what \dot{toe}_z^a represents. Is this the velocity of the contact leg, or the one of the swinging leg?  

7. It is not clear what is the difference between the "height in plane motion" and the "height in slope motion" mentioned in 210 and 211. 

8. The phrase in 215-216 is, in my opinion, unclear and counter-intuitive. Why increasing the stiffness of the feet will result in "fully contacting" the slope, while low stiffness will not? One would expect the opposite, as low stiffness makes the robot more flexible/deformable and thus more adaptable to the environment. 

9. The way the formulas are introduced are not following the known scientific conventions. As far as I'm concerned, the formulas can not be at the start of a sentence without any introduction/explanation before them. This is confusing for the reader. 

10. Fig. 3 and the simulation in general is not clear. The contact points between the robot and the ground seem to change at t=0.4s. Is this the case under the action of this controller? Why the ball is present again at t=1.2s? What do the magenta lines represent? The robot seems to move towards the opposite to the impact direction at t=0.6s. Is this related to an overshoot of the controller?

11. Based on the fact that the proposed control schemes are simple PD-like controllers and does not involve any adaptive online trajectory (gait) generation mechanism, it is not clear how the robot opted to lift its left foot at t=0.8s of Fig. 5 (i.e. to "step to adjust", as stated at line 279), in order to remain stable. Is this the expected behavior? If this is performed due to a given predefined/pre-planned trajectory tuned for the experiment or if the robot performs a periodic pre-defined gait motion, it should be clearly stated to avoid confusion.   

Reviewer 2 Report

This article dealt with the stability control strategy for a quadruped robot. The authors proposed a strategy using the quadratic program method to propose a system where the quadruped could regain its orginal state when subject to an external disturbance from the environment. Overall, the article is well explained and the supporting simulations have been provided. However, I prefer the authors to consider the following comments to the article in order to make it more interesting to the readers:

1. I suggest the authors to add a clear quality image of the robot under study and demonstrate the parameters they have analyzed on this image. This will allow readers to better correlate the analysis done

2. Figure 2 can be improved to a high quality image

3. Will it be possible to share a simulation video or the experiments link with the article?

4. Conclusion is too brief and I suggest the authors to add more information such as the disadvantages or improvements on the technique employed

Reviewer 3 Report

The manuscript is clearly written and easy to follow.

There are some minor issues to point out:

1) In lines 198-200, the authors wrote as follows:

"when the foot tip speed is zero but the force sensor has not yet judged the touchdown, it is necessary to give the foot tip a speed in the z direction to ensure stable contact between the foot end and the contact surface:"

The reviewer wonders if this can cause impact force when the foot hits the ground, which in turn cause instability.

2) It might be helpful if the authors could provide what data is delivered between the blocks in Fig. 1.

3) There is no explanation on theta 1 and theta 2 in the manuscript, while they appear in Fig. 2 (c).

4) In section 4, the authors might want to add experimental results on how stiffness is adjusted according to the environment.

5) Some of the plots in section 5 are difficult to read due the line color and thickness.

6) To further improve the quality of the manuscript, it needs minor corrections in English expressions.

Reviewer 4 Report

This is a well written paper supported with experimental datas.

- Graphical figures are so small, no need to put them horizontaly you can change them vertically . All figues under others as a one column. 

- Can you explain your simulation environment? Pyton, ros, c++, matlab, simscape? Which platform did you used.

Round 2

Reviewer 1 Report

The authors did a good job answering my comments. They also did the appropriate changes in the revised manuscript. I don't have any other comments.